# Exploring Avian Influenza Viruses in Yakutia—The Largest Breeding Habitat of Wild Migratory Birds in Northeastern Siberia

**DOI:** 10.3390/v17050632

**Published:** 2025-04-27

**Authors:** Nikita Kasianov, Kirill Sharshov, Anastasiya Derko, Ivan Sobolev, Nikita Dubovitskiy, Arina Loginova, Evgeniy Shemyakin, Maria Vladimirtseva, Nikolay Egorov, Viacheslav Gabyshev, Yujin Kim, Sun-Hak Lee, Andrew Y. Cho, Deok-Hwan Kim, Tae-Hyeon Kim, Chang-Seon Song, Hyesung Jeong, Weonhwa Jheong, Yoonjee Hong, Junki Mine, Yuko Uchida, Ryota Tsunekuni, Takehiko Saito, Alexander Shestopalov

**Affiliations:** 1Federal Research Center of Fundamental and Translational Medicine, Siberian Branch, Russian Academy of Sciences (FRC FTM SB RAS,), Novosibirsk 630060, Russia; nauka@duck.com (N.K.); a.derko19@gmail.com (A.D.); sobolev.riov@yandex.ru (I.S.); nikitadubovitskiy@gmail.com (N.D.); loginova995@gmail.com (A.L.); shestopalov2@mail.ru (A.S.); 2Institute of Biological Problems of the Cryolithozone, Siberian Branch, Russian Academy of Sciences, Yakutsk 677980, Russia; shemyakine@mail.ru (E.S.); sib-ykt@mail.ru (M.V.); epusilla@mail.ru (N.E.); gabvich@mail.ru (V.G.); 3Avian Diseases Laboratory, College of Veterinary Medicine, Konkuk University, Seoul 05029, Republic of Korea; yujinml@kribb.re.kr (Y.K.); cnescis@naver.com (S.-H.L.); 7rewcho@gmail.com (A.Y.C.); zh4697@naver.com (D.-H.K.); sakye12@naver.com (T.-H.K.); songcs@konkuk.ac.kr (C.-S.S.); 4Korea Research Institute of Bioscience and Biotechnology (KRIBB), Cheongju 28116, Republic of Korea; 5National Institute of Wildlife Disease Control and Prevention (NIWDC), Gwangju 61012, Republic of Korea; halley@korea.kr (H.J.); yoongjee@korea.kr (W.J.); 6National Institute of Biological Resources (NIBR), Incheon 22689, Republic of Korea; purify@korea.kr; 7Division of Transboundary Animal Disease, National Institute of Animal Health, Tsukuba 305-0856, Japan; minejun84032@affrc.go.jp (J.M.); uchiyu@affrc.go.jp (Y.U.); tune@affrc.go.jp (R.T.); taksaito@affrc.go.jp (T.S.)

**Keywords:** avian influenza, Yakutia, migratory birds, phylogenetics, surveillance, virus evolution

## Abstract

Yakutia, the largest breeding ground for wild migratory birds in Northeastern Siberia, plays a big role in the global ecology of avian influenza viruses (AIVs). In this study, we present the results of virological surveillance conducted between 2018 and 2023, analyzing 1970 cloacal swab samples collected from 56 bird species. We identified 74 AIVs of H3N6, H3N8, H4N6, H5N3, H7N7, H10N3, and H11N9 subtypes in Anseriformes order. Phylogenetic analysis showed that the isolates belong to the Eurasian lineage and have genetic similarities with strains from East Asia, Europe, and North America. Cluster analysis has demonstrated the circulation of stable AIV genotypes for several years. We assume that Yakutia is an important territory for viral exchange on the migratory routes of migrating birds. In addition, several amino acid substitutions have been found to be associated with increased virulence and adaptation to mammalian hosts, highlighting the potential risk of interspecific transmission. These results provide a critical insight into the ecology of the AIV and highlight the importance of continued monitoring in this geographically significant region.

## 1. Introduction

The influenza virus is one of the most well-known pathogens that has a significant impact on public health and the global economy. According to ICTV [1], the family Orthomyxoviridae includes the genera *Alphainfluenzavirus*, *Betainfluenzavirus*, *Deltainfluenzavirus*, and *Gammainfluenzavirus*. *Alphainfluenzavirus* is a pathogen of great epidemiological importance, capable of causing seasonal outbreaks and pandemics. Influenza A virus can infect a wide range of mammalian species, including humans, horses, domestic cattle, pigs, and seals, as well as avian species [2,3,4]. Among birds, it often causes epizootics, which result in severe economic losses to the poultry industry due to the mandatory culling of large numbers of farmed birds [5,6]. Wild waterfowl, as natural reservoirs of the influenza A virus, contribute to the spread of this pathogen over long distances, especially during seasonal migrations.

The Siberian region is of immense importance in light of the circulation of the influenza A virus. Here, the migratory routes of wild migratory birds intersect, and there are numerous rivers and lakes, which serve as breeding grounds for species that are ecologically linked to water bodies. The number of monitoring studies in various areas of Siberia varies, partly due to their inaccessibility. For example, the Republic of Sakha (Yakutia) is of great interest for studying the spread of the influenza A virus by wild waterfowl. Over 300 species of wild birds are found in the republic, 220 of which are migratory [7]. However, there are a limited amount of virological data available from this region. Influenza A viruses of various subtypes were isolated by researchers K. Okazaki et al. 2000 from duck fecal samples collected at breeding sites in Eastern Siberia, including Yakutia, from 1996 to 1998. Phylogenetic analysis of NP gene sequences of isolates collected during the study in Siberia and Hokkaido, Japan, showed that they belong to the Eurasian lineage of avian influenza viruses. It was also noted that the genes of the Siberian isolates are closely related to the genes of H5N1 influenza virus strains isolated from chickens and humans in Hong Kong in 1997, as well as to isolates from domestic birds in Southern China [8]. In 2014, researchers Marchenko et al. 2015 isolated a highly pathogenic influenza A virus strain A/wigeon/Sakha/1/2014, subtype H5N8, from the Republic of Sakha (Yakutia). Phylogenetic analysis of the HA and NA segment sequences indicated their affiliation with the Eurasian genetic lineage of avian influenza viruses. The authors concluded that the northern regions of the Russian Far East (particularly the territory of Yakutia) could serve as an intermediate site for the spread of the virus, based, in part, on the analysis of the chronology and nature of H5N8 outbreaks in Southeast Asia, Europe, and North America [9].

We analyzed 1970 cloacal swab samples from 56 bird species collected between 2018 and 2023 in the Republic of Sakha (Yakutia) as part of epidemiological surveillance activities. Our study significantly contributes to the global understanding of the spread of influenza A virus and provides insights into the role of Yakutia’s territory—one of the largest breeding sites of wild migratory birds in Northern Eurasia—in this process.

## 2. Materials and Methods

### 2.1. Ethical Issues

This study was conducted with the approval of the Biomedical Ethics Committee of FRC FTM SB RAS, Novosibirsk (Protocols No. 2019-3 and 2021-10). The bird specimens were collected during the hunting season (from 18 August to 15 October of each year), with a license from the regional Ministries of Ecology and Natural Resources, as part of the annual collection of biological material (i.e., the Program for the Study of Infectious Diseases of Wild Animals, FRC FTM, Novosibirsk).

### 2.2. Sample Collection

Cloacal swabs of freshly hunted wild waterfowl were collected during the official hunting season in individual 2 mL tubes filled with 1 mL of viral transport medium without glycerol, consisting of phosphate-buffered saline (PBS, pH 7.5), amphotericin B (15 µg/mL), penicillin G (100 units/mL), and streptomycin (50 µg/mL). The tubes were immediately stored in liquid nitrogen and were transported to the FRC FTM SB RAS laboratory for analysis [10]. The collection of material was undertaken from 2018 to 2023 in places of the highest density of bird populations in the Republic of Sakha (Yakutia).

### 2.3. Avian Influenza Virus Isolation Using Chicken Embryos

For the isolation of AIVs, an aliquot was taken for each sample, was mixed using a vortex shaker, and then the suspension was centrifuged for 3 min at 3000× *g*. Antibiotics (penicillin and gentamicin) were added to the supernatant, which was transferred to a new test tube to avoid bacterial infection. Next, 100 µL of each sample was inoculated into the allantoic cavity of two specific pathogen-free (SPF) chicken embryos and incubated at 37 °C for 72 h in our Biosafety level-3 (BSL-3) laboratory [11]. After cultivation, 2 mL of each allantoic fluid was extracted and used for a hemagglutination test (HA) with 5% chicken red blood cells [12]. All HA-positive samples were aliquoted for AIV M gene PCR testing. We conducted three consecutive inoculation passages in specific pathogen-free (SPF) chicken embryos. Samples demonstrating no hemagglutinating activity after these passages were considered negative.

### 2.4. RNA Extraction, Reverse Transcription, and PCR

All samples with HA activity were tested for the presence of influenza A. For this, RNA was extracted from the allantoic fluid using the RIBO-sorb kit (AmpliSens, Moscow, Russia). The resulting RNA was used in the reverse transcription reaction using the REVERTA-L kit (AmpliSens, Moscow, Russia). The presence of conserved M gene regions of the influenza A virus was determined using real-time PCR using the AmpliSens Influenza virus A/B-FL kit (AmpliSens, Moscow, Russia), developed to detect both human and avian influenza viruses.

### 2.5. Sequencing of AIVs

For complete genome sequencing of the viruses collected in 2018, RNA was isolated from allantoic fluids using the GeneJET viral DNA/RNA purification kit (Thermo Fisher Scientific, Waltham, MA, USA) and treated with TURBO DNase (Thermo Fisher Scientific). Up to 200 ng of RNA was used for preparation of the DNA libraries, using the NEBNext Ultra RNA Library Prep kit (New England Biolabs, Ipswich, MA, USA). Sequencing of the DNA libraries was conducted with a Reagent kit Version 3 (600-cycle) using the MiSeq genome sequencer (Illumina, San Diego, CA, USA) in the Genomics Core Facility (ICBFM SB RAS, Novosibirsk, Russia). Full-length genomes were assembled de novo with CLC Genomics Workbench v.9.5.3 (Qiagen, Hilden, Germany). The raw data of the sequences that were used for assembly have not been deposited and are stored and available in the “Genomics Core Facility” (ICBFM SB RAS, Novosibirsk, Russia).

Viruses isolated in 2019–2021 were sequenced as part of a joint international program with the National Institute of Animal Health (Tsukuba, Japan). Thus, isolation of RNA from allantoic fluid was carried out using the RNeasy Mini kit (QIAGEN, Hilden, Germany). We used the NEBNext Ultra II RNA Library Prep Kit for Il-lumina (New England Biolabs, Ipswich, MA, USA) to prepare cDNA libraries. In total, 10 pM of libraries was mixed with 10 pM of PhiX Control V3 (Illumina, San Diego, CA, USA) before sequencing. Sequencing was performed using the MiSeq genome sequencer (Illumina), using MiSeq Reagent Kit v.2 (Illumina). Consensus sequences were constructed using Workbench software v.9.5.3 (QIAGEN, Germany). The most relevant reference sequences from the U.S. National Center for Biotechnology Information (NCBI) GenBank database were included in the analysis.

Viruses isolated in 2022–2023 were sequenced using Illumina MiSeq platform and associated reagent kits, also from Illumina, according to the manufacturer’s methodology. RNA was extracted using the QIAamp Viral RNA Mini Kit (Qiagen, Germany). Whole-genome amplification was performed using the modified protocol presented by Bin Zhou [13]. DNA libraries were prepared using a Nextera DNA Flex Library Prep kit (Illumina). Sequencing of the DNA libraries was conducted with a reagent kit, version 3 (600-cycle), on a MiSeq genome sequencer (Illumina). Raw Illumina sequencing reads were analyzed and assembled using the CFIA-NCFAD/nf-flu workflow (v3.2.1) [14], implemented on the Nextflow platform (v. 24.04.2.5914) [15].

### 2.6. Search for the Nearest Identical Sequences for AIVs

For each of the sequences of the segments of influenza viruses obtained, a search was carried out for identical or the most similar sequences using the local version BLAST+ (v2.16.0) [16] from the GISAID database [17].

### 2.7. Phylogenetic and Cluster Analyses of AIVs

The sequence sets were subjected to multiple alignment using MAFFT v7.520 multiple alignment algorithm (downloadable version) [18] using the default parameters, as follows: gap opening penalty: 1.53 and gap extension penalty: 0.0. Files containing multiple alignments were then opened in the Unipro UGENE program, version 50.0 [19], where sites containing deletions were manually checked and sites containing deletions were removed from the final alignment.

Using IQ-TREE multicore version 2.3.6 for Linux x86 64-bit, built 4 Aug 2024 [20], phylogenetic trees were constructed using the maximum likelihood (ML) method and branch support was assessed using bootstrap with the number of iterations being equal to 500. For each gene segment, the model of nucleotide substitutions was determined using ModelFinder through IQ-TREE version 1.6 [21].

Manipulations with files containing sequences and tables were performed using the Python programming language and its libraries, such as Biopython 1.81 [22].

Clusters were identified and marked using Cluster Picker v1.2.3 [23], using thresholds of 0.045 of maximum pairwise genetic distance between sequences and a branch support of 90.

The construction of phylogenetic trees and the identification of clusters for the internal gene sequences of the studied strains were performed without adding other sequences to the analysis. For the HA and NA segments, sequences from the GISAID database found via BLAST were added to the phylogenetic and clustering analyses.

Tree visualization and topology analyses were performed in iTOL v.6 [24].

### 2.8. Search for Amino Acid Substitutions in Internal Genes of AIVs

We analyzed the proteins of the internal genes of our samples for the presence of amino acid substitutions described earlier [25]. Search for amino acid substitutions in internal genes was carried out in Unipro UGENE program.

## 3. Results

### 3.1. Ecological Features of Study Area and Samples

Within the framework of this project, samples were collected from wild birds of nine orders: *Accipitriformes*, *Anseriformes*, *Charadriiformes*, *Galliformes*, *Gaviiformes*, *Gruiformes*, *Passeriformes*, *Pelecaniformes*, and *Podicipediformes*. From 2018 to 2023, samples were collected in nine regions of Yakutia (Figure 1). The total number of samples collected was 1970. The majority of the studied birds belonged to the order *Anseriformes*. The proportion of individuals from this order amounted to 93.96% of the total number of collected samples. The remaining eight orders accounted for 6.04% of the total number of studied birds. Of the 1851 representatives of the order *Anseriformes,* several species made up a significant portion of the studied individuals: mallard (*Anas platyrhynchos*, n = 220), common teal (*Anas crecca*, n = 516), northern shoveler (*Anas clypeata*, n = 174), tufted duck (*Aythya fuligula*, n = 250), and tundra swan (*Cygnus bewickii*, n = 218) (Table 1). The information of distribution of bird species by year, sample information, time, and location is provided in Appendix A.

### 3.2. AIVs in Wild Birds of Yakutia

Between 2018 and 2023, during virological monitoring in the territory of Yakutia, 74 influenza A virus strains were isolated from the 1970 collected samples (Table 2). All of them were isolated from birds belonging to the order *Anseriformes.* Among the common teal (*Anas crecca*), the number of positive samples was 36, for the northern shoveler (*Anas clypeata*)—16, for the mallard (*Anas platyrhynchos*)—8, for the garganey (*Anas querquedula*) —6, for the northern pintail (*Anas acuta*)—3, and for the falcated duck (*Anas falcata*), baikal teal (*Anas formosa*), European wigeon (*Anas penelope*), and tufted duck (*Aythya fuligula*)—1 sample each.

### 3.3. Cluster Analysis and the Nearest Sequences from the GISAID Database for the Internal Genes of AIVs

A table (Table 3) was created, in which the affiliation of each sample’s internal segments to a specific cluster is reflected by color, and the sampling location of nearest sequence by percent identity from the GISAID database is also indicated.

The cluster sets for the internal genes of the studied AIV strains were conditionally named Genotype1 (g1) to Genotype9 (g9). G1 includes the strains A/Common Teal/Yakutia/14/2019, A/Common Teal/Yakutia/32/2019, A/Shoveler/Yakutia/56/2019, A/Shoveler/Yakutia/61/2019, and A/Teal/Yakutia/849/2018. It is interesting to note that this group includes samples collected in both 2019 and 2018. The closest identical sequences from the GISAID database for this group of studied strains show geographical homogeneity within each segment and primarily come from different Asian countries. However, for the PB2 segment, the closest sequences come from Egypt and the Novosibirsk region of Russia for the strain A/Teal/Yakutia/849/2018.

G2 includes the strains A/Anas Crecca/Yakutia/K-57/2023, A/Garganey/Yakutia/K-74/2023, and A/Shoveler/Yakutia/K-33/2023. The sequences of the internal segments of these samples show their identity to viruses from different Asian countries.

G3 includes the strains A/Common Teal/Yakutia/K-28/2023, A/Common Teal/Yakutia/K-96/2023, A/Common Teal/Yakutia/K-98/2023, A/Northern Shoveler/Yakutia/K-56/2023, A/Shoveler/Yakutia/Bo17/2022, A/Shoveler/Yakutia/Bo29/2022, A/Shoveler/Yakutia/K-17/2023, A/Shoveler/Yakutia/K-68/2023, and A/Shoveler/Yakutia/K-72/2023. Also, it can be seen that this group of strains includes samples collected in both 2023 and 2022. The closest identity samples for the studied strains mostly come from countries in East and Southeast Asia.

The strains A/Baikal Teal/Yakutia/OP-16/2023 and A/Tufted Duck/Yakutia/K-37/2023 make up G4. For all segments of these strains, the closest identical sequences come from Japan and Korea.

G5 consists of the strains A/Common Teal/Yakutia/80*/2021 and A/Shoveler/Yakutia/141*/2021. The sequences of the internal segments of these samples show their identity to viruses from China, Korea, Japan, and, for the NS segment, the strain A/Common Teal/Yakutia/80*/2021 is most identical to the virus isolated in Alaska.

G6 includes the strains A/Common Teal/Yakutia/C-74/2023, A/Common Teal/Yakutia/C-86/2023, and A/Garganey/Yakutia/K-75/2023. The closest identity samples for the internal segments of these strains mostly come from different Asian countries.

G7 consists of the samples A/Common Teal/Yakutia/128/2020, A/Common Teal/Yakutia/49/2020, A/Common Teal/Yakutia/57x/2020, A/Common Teal/Yakutia/59x/2020, A/Common Teal/Yakutia/60x/2020, A/European Wigeon/Yakutia/55/2020, A/Garganey/Yakutia/94/2020, A/Garganey/Yakutia/98/2020, A/Mallard/Yakutia/11/2020, A/Mallard/Yakutia/21/2020, A/Northern Pintail/Yakutia/14/2020, A/Shoveler/Yakutia/25/2020, and A/Shoveler/Yakutia/65/2020. The closest identical sequences from the GISAID database for this group of studied strains show their geographical homogeneity within each segment and come from regions such as Mongolia, the Amur region of the Russian Federation, Japan, Korea, and China.

G8 includes the strains A/Common Teal/Yakutia/19/2019, A/Shoveler/Yakutia/57/2019, and A/Shoveler/Yakutia/68x/2019. The closest identical sequences for this group of studied strains show their geographical homogeneity within each segment and come from Mongolia, the Amur region of the Russian Federation, Korea, and China.

The strains A/Common Teal/Yakutia/6x/2019 and A/Shoveler/Yakutia/28x/2019 make up G9. The sequences of the internal segments of these samples show their identity to viruses from the same countries as those in G8.

The presence of strains collected in different years within the same group may indicate the circulation of one set of internal genes of the influenza virus over time.

Some studied strains, such as A/Teal/Yakutia/802/2018, A/Common Teal/Yakutia/148*/2021, A/Mallard/Yakutia/47/2020, A/Common Teal/Yakutia/UA-4/2023, A/Northern Pintail/Yakutia/C-82/2023, A/Common Teal/Yakutia/C-83/2023, A/Common Teal/Yakutia/2x/2019, A/Mallard/Yakutia/C-3/2023, A/Common Teal/Yakutia/UA-3/2023, A/Common Teal/Yakutia/UA-52/2023, A/Common Teal/Yakutia/V98/2022, A/Mallard/Yakutia/K-52/2023, A/Mallard/Yakutia/179*/2021, A/Common Teal/Yakutia/18/2019, A/Common Teal/Yakutia/122*/2021, A/Common Teal/Yakutia/V109/2022, A/Falcated Duck/Yakutia/OP-62/2023, A/MallardandNorthern Pintail Hybrid/Yakutia/V46/2022, A/Common Teal/Yakutia/13*/2021, A/Common Teal/Yakutia/C-52/2023, A/Common Teal/Yakutia/111*/2021, A/Common Teal/Yakutia/UA-55/2023, A/Shoveler/Yakutia/Bo30/2022, A/Common Teal/Yakutia/UA-37/2023, A/Common Teal/Yakutia/18*/2021, A/Common Teal/Yakutia/63x/2019, A/Northern Pintail/Yakutia/41/2019, and A/Mallard/Yakutia/178*/2021, did not form unique groups with other strains. However, in this table, we can observe the proximity of some strains by individual genomic segments. For example, the strains A/Common Teal/Yakutia/122*/2021, A/Common Teal/Yakutia/V109/2022, A/Falcated Duck/Yakutia/OP-62/2023, A/MallardandNorthern Pintail Hybrid/Yakutia/V46/2022, A/Common Teal/Yakutia/13*/2021, A/Common Teal/Yakutia/C-52/2023, A/Common Teal/Yakutia/111*/2021, and A/Common Teal/Yakutia/UA-55/2023 have a PB2 segment that is closely related.

The NS segment sequences of the strains A/Common Teal/Yakutia/2x/2019 and A/Common Teal/Yakutia/V98/2022 are closest to strains from Germany and South Africa, respectively, while the MP segment sequence of the strain A/Common Teal/Yakutia/122*/2021 is most closely related to a strain isolated in Georgia. This stands out against the background of the closeness of the other sequences to “Asian” ones.

Additionally, the sequences of some segments were not clustered. For example, the viruses A/Common Teal/Yakutia/146*/2021, A/Common Teal/Yakutia/C-67/2023, A/Common Teal/Yakutia/C-85/2023, and A/Mallard/Yakutia/C-9/2023 form a single group for the PB2, PB1, PA, NP, and MP segments, but the NS segment sequences of these strains did not fall into any cluster. Looking at their placement on the phylogenetic tree, it can be seen that they are located in different clades.

It is also interesting to note that for the MP segment, the strain A/Common Teal/Yakutia/V109/2022 is most closely related to a virus isolated from a human in China.

### 3.4. Amino Acid Substitutions in the Proteins of the Internal Genes of AIVs

The protein sequences of the internal segments of the studied strains were analyzed for significant amino acid substitutions. A total of 37 substitutions were examined for the PB2 protein, 12 substitutions for the PB1 protein, 2 substitutions for the PB1-F2 protein, 23 substitutions for the PA protein, 13 substitutions for the NP protein, 19 substitutions for the NS1 protein, 4 substitutions for the NS2/NEP protein, 3 substitutions for the M1 protein, and 5 substitutions for the M2 protein. Information on all the examined amino acid positions, substitutions, and their effects is presented in Appendix A . The identified amino acid substitutions are presented in Table 4.

### 3.5. Phylogenetic and Cluster Analyses of AIVs

Phylogenetic trees for all genome segment sequences of the studied strains were constructed using the maximum likelihood method with bootstrap support for the branches. Here, we present dendrograms for the HA and NA segments. Phylogenetic trees with designated clusters for the PB2, PB1, PA, NP, MP, and NS (internal segments) segments are provided in the Appendix A.

The studied strains with the hemagglutinin subtype H3, based on the HA segment, belong to the Eurasian genetic lineage of avian influenza viruses (Figure 2). For the most part, they show their kinship with samples isolated from various parts of Asia, predominantly from its eastern and southeastern regions. However, the sequences of the strains A/Garganey/Yakutia/K-74/2023 and A/Anas Crecca/Yakutia/K-57/2023 form a clade with viruses isolated from the European part of the continent, Central Asia, and Western Siberia.

Based on the NA segment, the studied strains of subtype N8 also belong to the Eurasian genetic lineage of avian influenza viruses (Figure 3). It is worth noting that the strains A/Garganey/Yakutia/K-74/2023 and A/Anas Crecca/Yakutia/K-57/2023 show their kinship with samples from East Asia, but they are also distanced from the other studied strains. We can observe several large clades on the phylogenetic tree, in which multiple clusters merge. It is interesting to highlight the clade containing the strains A/Common Teal/Yakutia/UA-4/2023, A/Northern Pintail/Yakutia/C-82/2023, and A/Common Teal/Yakutia/C-83/2023. This clade also includes a sample collected in the Yakutia Region in 2014, which forms, in turn, one cluster and clade with strains from Japan, Korea, Hungary, and Germany. Additionally, the dendrogram contains strains collected in Alaska. The Yakutian viruses within the clade named East Asian-like clade 2 show the closest kinship to them.

The studied strains with the neuraminidase subtype N6 also belong to the Eurasian genetic lineage of avian influenza viruses (Figure 4). Our cluster analysis shows that the strains form two clades on the phylogenetic tree. The strain A/Teal/Yakutia/802/2018 is located in a separate clade and shows its kinship with viruses from Korea and the Russian Far East. Although it itself remains unclustered, some other strains in this clade belong to a separate cluster (Clust2). The strain A/Common Teal/Yakutia/148*/2021 is also located in a separate clade. This clade consists of samples from Mongolia, Buryatia, and China (Beijing). Additionally, the strain A/Common Teal/Yakutia/V109/2022 forms a single clade with viruses collected in East Asia.

For the HA segment, the studied strains with the H4 subtype also belong to the Eurasian genetic lineage of avian influenza viruses (Figure 5). Cluster analysis revealed two clades on the phylogenetic tree; however, all studied samples, except A/Common Teal/Yakutia/V109/2022, belong to the same cluster (Clust1). The strain A/Common Teal/Yakutia/V109/2022 forms a clade with viruses collected from various parts of the Eurasian continent, from Europe to East Asia. The remaining studied strains show their kinship primarily with samples from East Asia.

The only strain with the H7N7 subtype is A/Mallard/Yakutia/47/2020. The sequences of its external segments also belong to the Eurasian genetic lineage of avian influenza viruses (Figure 6). For the HA segment, this strain shows its kinship with viruses collected from East Asia, although the phylogenetic tree also includes strains from Egypt, Georgia, and Dagestan, all of which belong to the same cluster (Clust1). The NA segment sequence of the studied strain A/Mallard/Yakutia/47/2020 is also closely related to strains collected from East Asia.

The H11N9 subtype is represented by two studied strains: A/Baikal Teal/Yakutia/OP-16/2023 and A/Tufted Duck/Yakutia/K-37/2023. The sequences of the HA and NA segments of these strains belong to the Eurasian genetic lineage of avian influenza viruses (Figure 7). Both segments show relatedness to viruses primarily collected from countries in East Asia.

It is also interesting to note that cluster and phylogenetic analysis of the internal segments of the strains A/Baikal Teal/Yakutia/OP-16/2023 and A/Tufted Duck/Yakutia/K-37/2023 indicated their belonging to the same group, as mentioned earlier. It is worth noting that these viruses were isolated in different regions of Yakutia (Tomponsky and Namsky districts, respectively) and in different months (September and August, respectively).

The H10N3 subtype is represented by the studied strains A/Falcated Duck/Yakutia/OP-62/2023 and A/Common Teal/Yakutia/2x/2019. In the phylogenetic tree of the HA segment, they are found in different clades and clusters (Figure 8). The HA segment sequence of the strain A/Common Teal/Yakutia/2x/2019, although unclustered, shows relatedness to viruses collected in the Novosibirsk region and the Netherlands (forming Clust3). Additionally, in this clade, we can observe viruses from the European part of the continent (Denmark, Sweden, Georgia). For this segment, the studied strain belongs to the Eurasian genetic lineage of avian influenza viruses.

The clade that includes the strain A/Falcated Duck/Yakutia/OP-62/2023 is formed by sequences from viruses isolated in North America and the Asian part of Eurasia (Figure 8). The studied strain A/Falcated Duck/Yakutia/OP-62/2023 belongs to the American genetic lineage of avian influenza viruses based on its hemagglutinin segment. The HA segment sequence of the studied strain A/Falcated Duck/Yakutia/OP-62/2023 is most closely related to a virus isolated from a human in China in 2023. Also, as mentioned earlier, the MP segment sequence of the studied virus A/Common Teal/Yakutia/V109/2022, which has the subtype H4N6, is closely related to this strain.

For the NA segment, both strains A/Falcated Duck/Yakutia/OP-62/2023 and A/Common Teal/Yakutia/2x/2019 belong to the Eurasian genetic lineage of avian influenza viruses (Figure 8). A/Common Teal/Yakutia/2x/2019 is in a separate clade formed by two clusters (Clust1 and Clust2), showing the greatest relatedness to a virus isolated in Dagestan. Another large clade formed by Clust3 contains strains from different countries in Europe and Asia. It includes the strain A/Falcated Duck/Yakutia/OP-62/2023, which is closely related to a virus isolated in Bangladesh, as well as another studied strain, A/Mallard/Yakutia/C-3/2023.

The H5N3 subtype has a single strain, A/Mallard/Yakutia/C-3/2023, and based on its HA segment, it shows relatedness to viruses predominantly isolated in East Asia, such as in Japan.

## 4. Discussion

The territory of Yakutia, located in the northeastern part of Russia, is an important point on the migratory routes of many species of migratory birds that use this area for seasonal migration. The abundance of water and the availability of food are the primary factors influencing this. Many bird species migrating through Yakutia are rare and protected, including at the global level. Since Yakutia lies on the route between the Arctic and the more southern regions of Asia, Europe, and even America, its natural conditions provide suitable places for temporary stops and nesting, especially along major rivers and lakes.

We suggest Yakutia is a key region for studying the mechanisms of transmission and evolution of avian influenza due to its unique combination of ecological, geographical, and climatic factors [9]. The territory of Yakutia encompasses Arctic tundras, northern and middle taiga, mountain systems (e.g., the Verkhoyansk Range), and extensive water systems, including the Lena, Yana, Indigirka, and Kolyma rivers, as well as thousands of lakes [37]. These ecosystems create ideal conditions for nesting, migration, and stopovers of waterfowl and shorebirds (ducks, geese, sandpipers), which are the primary reservoirs of avian influenza virus. The high density of water bodies facilitates bird congregation, increasing the risk of pathogen exchange between species and populations [38].

The region lies at the intersection of four major migratory flyways: the West Asian–East African Flyway, the Central Asian Flyway, the West Pacific Flyway, and the East Asian–Australasian Flyway (EAAF). According to ringing data and modern tracking studies (e.g., GPS transmitters and geolocators), birds nesting in Yakutia migrate to wintering grounds in North America, Africa, Southeast Asia, Australia, and New Zealand. This positions Yakutia as a “bridge” between continents, where viruses circulating in isolated populations can mix, evolve, and spread globally [39,40,41,42,43,44].

Yakutia’s sharply continental climate, with extreme temperature fluctuations, influences virus survival in the environment. Low winter temperatures prolong pathogen persistence in water and soil, increasing transmission risks during migrations. Conversely, the short but intense nesting season in summer creates conditions for rapid viral spread among dense bird aggregations [45,46].

Despite fragmented data on migratory links (e.g., ringing records of birds from Central Yakutia wintering in China, Japan, and India, or those from Northeast Yakutia in North America), the region’s precise role in transcontinental virus transmission remains understudied. Modern tracking and genomic analysis methods can clarify migration corridors, population contact points, and outbreak “hotspots”, which are critical for predicting pandemic risks.

Several migratory routes of wild migratory birds intersect in Yakutia. The main ones are the Central Asian route and the East Asian-Australian route. Some species also use east-west migratory routes. The Central Asian route covers vast inland areas between the Arctic and the Indian Oceans, while the East Asian-Australian migratory route stretches from the Russian Far East and the Alaska Peninsula, connecting Eurasia and North America in the north, and Australia and New Zealand in the south.

By surveying 1970 bird specimens of 56 species, we obtained complete genomes of 74 viruses. Further phylogenetic and clustering analyses allowed us to closely study possible routes of avian influenza virus transmission with migratory birds in Yakutia.

We conducted a study of the obtained sequences for significant amino acid substitutions and found some strains with these substitutions. Some substitutions, such as N66S in the PB1-F2 protein, were present in a large number of strains, while others, like S224P, N383D in the PA protein, were found in only one. Monitoring the amino acid substitutions in this area, for which studies have shown various effects, such as increased virulence, is of great importance in the context of the ornithological significance of the Yakutia Region.

Summarizing the results of phylogenetic and clustering analyses of the internal segments, we concluded that avian influenza virus strains isolated in Yakutia are genetically distinct from one another and form clusters of phylogenetically related strains. Searching for the closest identical sequences revealed a relationship with viruses predominantly of “Asian” origin. Many of the closest sequences come from East Asian countries, such as Japan. Additionally, the most identical sequences were found from Alaska. All of this corresponds to the migratory flight routes passing through Yakutia. In the three cases where the greatest identity was observed with “European” viruses, and thus similarity with sequences isolated in regions corresponding to latitudinal distribution, as well as some differences in the phylogenetic surrounding of different segments of the same influenza virus strains, reassortment events may have occurred, leading to the emergence of virus variants with new segment combinations.

The epidemiological connection of Yakutia, particularly with East Asian territories, is also confirmed by some studies. For example, J. Mine et al. showed the relationship between the PA segment of highly pathogenic avian influenza A viruses of subtype H5N8, responsible for outbreaks in domestic and wild birds in Japan in 2020–2021, and viruses isolated in Yakutia, the Amur region, and Primorye [47]. The strains A/Common Teal/Yakutia/128/2020, A/Common Teal/Yakutia/49/2020, A/Common Teal/Yakutia/57x/2020, A/Common Teal/Yakutia/59x/2020, A/Common Teal/Yakutia/60x/2020, A/European Wigeon/Yakutia/55/2020, A/Garganey/Yakutia/94/2020, A/Garganey/Yakutia/98/2020, A/Mallard/Yakutia/11/2020, A/Mallard/Yakutia/21/2020, A/Northern Pintail/Yakutia/14/2020, A/Shoveler/Yakutia/25/2020, and A/Shoveler/Yakutia/65/2020 are also included in our study. They form a group by internal segments, named G7. For the PA segment, the closest strains to them are from the Amur region of Russia, which is consistent with the results of the aforementioned study.

There have been extensive large-scale studies of avian influenza virus along the East Asian–Australasian Flyway, which are highly relevant when considering collaborations and evaluating the current status and importance of AIV research and surveillance in Yakutia. Our study confirmed that the most common subtypes of low-pathogenic influenza viruses in wild birds in Yakutia are H3N8 and H3N6. H3Nx viruses are known to be widespread in wild bird populations in southeastern regions. For example, long-term research in Eastern China has demonstrated that H3N8 viruses circulating among migratory birds and ducks, comprising 12 genotypes, have evolved into different branches and undergone complex reassortment with viruses in waterfowl [48].

Phylogenetic analysis in our study showed that most Yakut H3 viruses cluster with viruses isolated from various parts of Asia, predominantly from Eastern and Southeastern China, forming two East Asian clades. In contrast, only six H3 viruses from Yakutia belong to Central Asian clades, and just two to European ones. This suggests a closer phylogenetic relationship between Yakut and East Chinese viruses and indicates the possibility of intensive segment exchange between these regions.

Recent human infections with H3N8 viruses in China have heightened interest in, and concern about, the evolution and cross-species transmission risk of these viruses. Therefore, it is advisable to conduct large-scale comparative studies of H3Nx viruses migrating between regions such as Yakutia in the north and Eastern China and other wintering areas of wild migratory birds [48]. Our study also identified some strains that are less common as well as their interesting relationship with viruses that have already affected humans. Specifically, the strain A/Falcated Duck/Yakutia/OP-62/2023, which has the H10N3 subtype, and its phylogenetic relationship with the hemagglutinin segment of the virus A/Zhejiang/CNIC-ZJU01/2023. The strain A/Zhejiang/CNIC-ZJU01/2023 | A/H10N5 | HA | 2023-12-15 | EPI ISL 18, 846, 022 was isolated in Zhejiang Province, China, in December 2023 from a 63-year-old woman who exhibited severe symptoms, such as high fever, cough, chest pain, and worsened by pre-existing chronic conditions. She was diagnosed with type I respiratory failure and severe pneumonia and ultimately passed away on 16 December 2023. However, in this case, coinfection with H3N2 and H10N5 strains was observed. Furthermore, the H10N5 virus was also isolated from frozen domestic duck meat stored in a refrigerator, which had been slaughtered earlier in November of the same year. Studies revealed that the hemagglutinin sequence belonged to the American genetic lineage and showed identity to the strain A/Anser albifrons/South Korea/22JN-163-1/2022, which may indicate an event of reassortment and interspecies transmission of the virus [49]. In our case, we can also observe the proximity of the investigated strain A/Falcated Duck/Yakutia/OP-62/2023 to a virus from Korea.

Phylogenetic analyses indicate that some H10Nx viruses from Eastern China cluster within the North American lineage and have established a novel Eurasian branch in wild birds across South Korea, Bangladesh, and China [50]. We can assume that such viruses have the potential to spread to Yakutia. However, our phylogenetic analysis shows that the H10N5 virus we identified belongs to the European clade, most likely originating from Western Eurasia and spreading eastward through several migration routes connecting Siberia and Europe [51].

We also identified a relatively rare subtype, H11N9, in Baikal teal and tufted ducks. Waterfowl and near-water birds are considered the main reservoirs of H11 viruses. Based on the literature data, this subtype has low prevalence in Eastern China, with an infection rate of approximately 0.067% [52]. Nonetheless, H11 viruses are susceptible to complex reassortment with circulating viruses among waterfowl and aquatic birds. Our viruses showed phylogenetic proximity of both H11 and N9 segments to Japanese strains, indicating a likely connection with eastward migration.

Additionally, there is experimental evidence suggesting that the H11Nx subtype has a high potential for spreading to domestic birds and mammals, highlighting its zoonotic risk [52].

The H7N7 subtype found in wild birds poses a significant threat to the poultry industry, particularly along the East Asia–Australia migration route. This underscores the importance of monitoring influenza viruses in both wild and domestic birds. Various clades of low-pathogenic H7Nx subtypes of European and Central Asian origin have been identified in Eastern China; these subtypes have the potential to become highly pathogenic [53]. Our study showed that the Yakut H7N7 virus is phylogenetically similar to viruses from Japan and Shandong, suggesting a connection through wild bird migrations. Therefore, comparative studies like these are essential for developing effective control strategies against the H7N7 virus.

We identified amino acid substitutions associated with increased virulence and adaptation in mammals. These substitutions can influence viral behavior and the potential risk of interspecies transmission, highlighting their prevalence in AIVs and their relevance to mammalian adaptation. However, given the environmental context and the minimal likelihood of contact with domestic animals and humans, we consider the potential public health impact to be low. Nevertheless, the detection of novel viruses and their amino acid substitutions in natural populations of wild migratory birds is a critical finding for the early warning efforts.

Clearly, the H7, H10, and H11 subtypes identified in this study should be compared in future research with those found in southeastern regions. Such comparisons will offer a broader perspective on the circulation of avian influenza viruses along the East Asia–Australasian flyway, including areas like Yakutia and Eastern China, and will enhance our understanding of AIV transmission and evolution across different regions and hosts.

In conclusion, it is important to emphasize the need for further research in the territory of Yakutia. Due to their small number, these studies are crucial for monitoring and forecasting the epidemiological situation as well as for the significance and uniqueness of Yakutia itself in ornithological and, accordingly, epidemiological terms.

## 5. Limitations of the Study

A limitation of the study is that the biological samples were collected exclusively during the permitted period of official hunting for waterfowl, regulated by the regional government of the Republic of Yakutia. It means the distribution of samples across seasons is limited. Therefore, we can only speculate about a temporary cut in the circulation during the autumn period of the beginning of bird migration to wintering grounds. It does not allow us to test any hypothesis seasonal patterns and how the seasonality influences virus detection rates.

Additionally, the study includes 93.96% of samples from Anseriformes, which may introduce bias. However, we have a focus on this major reservoir of AIVs to assess general diversity of viruses.

We have intentionally avoided a detailed analysis of amino acid substitutions, as such interpretations are not supported by our experimental data and were beyond the scope of this study. We did not conduct experiments specifically focused on substitution effects, including reverse genetics, which are necessary for such investigations. Instead, we believe that our ecological and phylogenetic analyses provide valuable and important insights for this research area.

## 6. Conclusions

This study highlights the critical role of Yakutia as a major breeding and migratory hub for wild birds, contributing to the spread and evolution of AIVs. Yakutia serves as a natural laboratory for studying the interplay between ecology, bird migration, and the evolution of zoonotic pathogens. Its geographic location, landscape diversity, and role in global migrations make it pivotal for understanding avian influenza spread and developing biosecurity strategies.

Our multi-year surveillance identified a diverse range of AIV subtypes, primarily within the Eurasian lineage, with phylogenetic links to strains from East Asia, Europe, and North America. The detection of conserved viral genotypes over multiple years suggests the persistence of certain AIV strains within this region, emphasizing its significance in global influenza ecology. Moreover, the presence of amino acid substitutions associated with increased virulence and mammalian adaptation raises concerns regarding potential interspecies transmission risks. These findings reinforce the necessity of continuous monitoring and international collaboration to assess the dynamics of AIV circulation and mitigate potential zoonotic threats. Further research should focus on expanding genomic analysis and evaluating the ecological factors influencing virus persistence in this vast and remote environment.

## Figures and Tables

**Figure 1 viruses-17-00632-f001:**
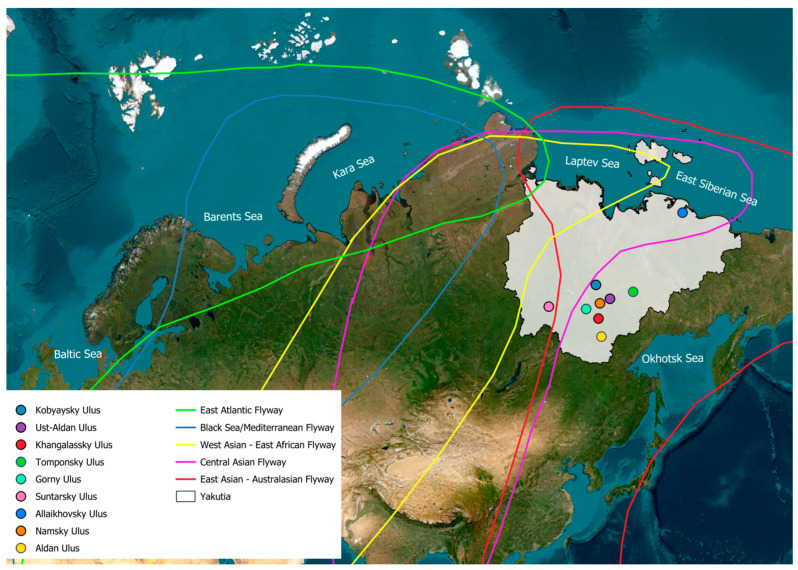
Sample collection areas in Yakutia Region of Eastern Siberia.

**Figure 2 viruses-17-00632-f002:**
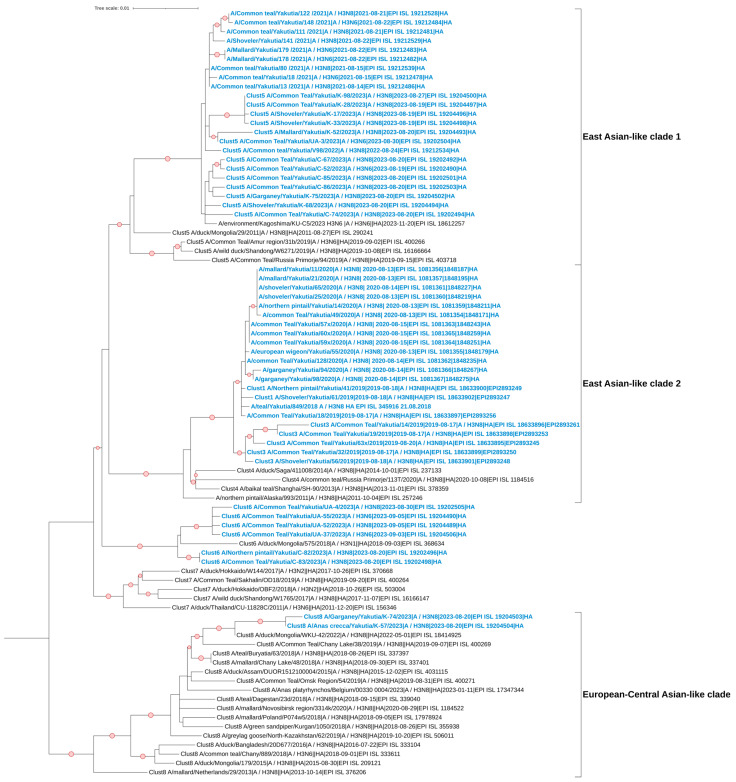
Maximum likelihood phylogenetic tree of the HA (H3) genome segment of avian influenza viruses isolated in the Republic of Sakha (Yakutia). The red circle symbol denotes branches with values bootstrap > 70%. The studied strains collected during this research are highlighted in blue. The tree scale represents the number of substitutions per site.

**Figure 3 viruses-17-00632-f003:**
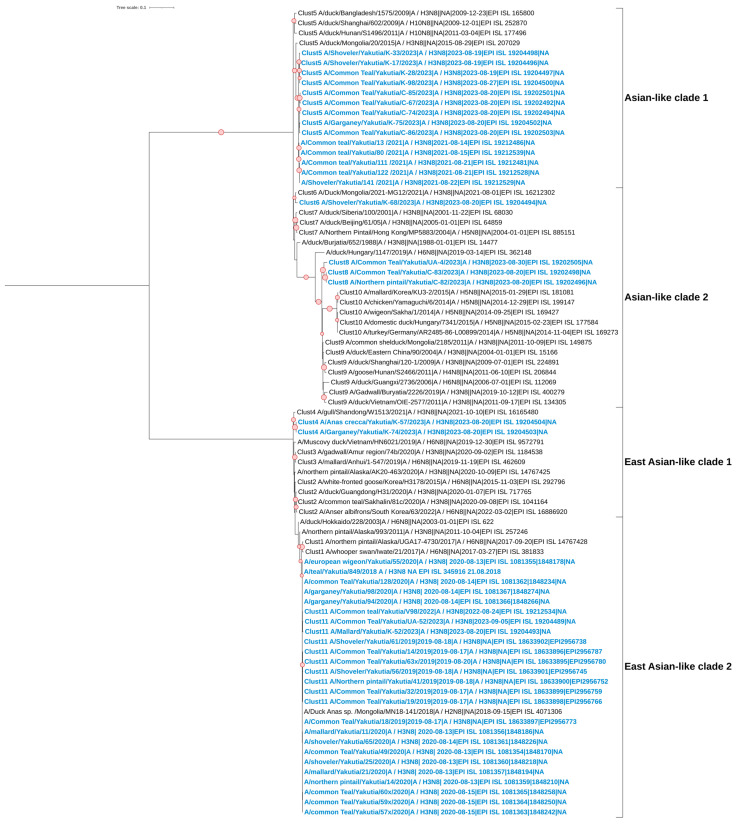
Maximum likelihood phylogenetic tree of the NA (N8) genome segment of avian influenza viruses isolated in the Republic of Sakha (Yakutia). The red circle symbol denotes branches with values bootstrap > 70%. The studied strains collected during this research are highlighted in blue. The tree scale represents the number of substitutions per site.

**Figure 4 viruses-17-00632-f004:**
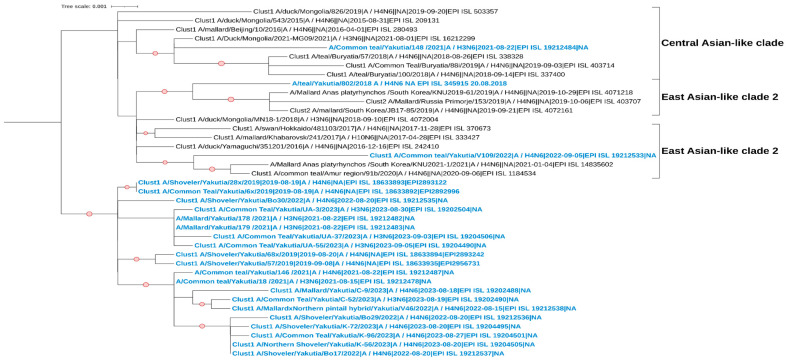
Maximum likelihood phylogenetic tree of the NA (N6) genome segment of avian influenza viruses isolated in the Republic of Sakha (Yakutia). The red circle symbol denotes branches with values bootstrap > 70%. The studied strains collected during this research are highlighted in blue. The tree scale represents the number of substitutions per site.

**Figure 5 viruses-17-00632-f005:**
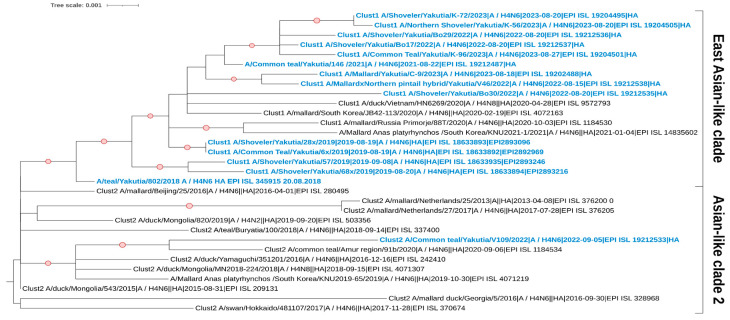
Maximum likelihood phylogenetic tree of the HA (H4) genome segment of avian influenza viruses isolated in the Republic of Sakha (Yakutia). The red circle symbol denotes branches with values bootstrap > 70%. The studied strains collected during this research are highlighted in blue. The tree scale represents the number of substitutions per site.

**Figure 6 viruses-17-00632-f006:**
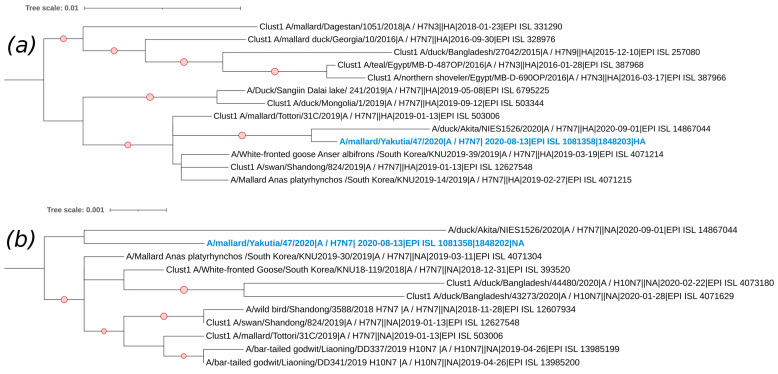
Phylogenetic analysis of H7N7 avian influenza viruses. (**a**) Maximum likelihood phylogenetic tree of the HA (H7) genome segment of avian influenza viruses isolated in the Republic of Sakha (Yakutia). The red circle symbol denotes branches with values bootstrap > 70%. The studied strains collected during this research are highlighted in blue. The tree scale represents the number of substitutions per site. (**b**) Maximum likelihood phylogenetic tree of the NA (N7) genome segment of avian influenza viruses isolated in the Republic of Sakha (Yakutia). The red circle symbol denotes branches with values bootstrap > 70%. The studied strains collected during this research are highlighted in blue. The tree scale represents the number of substitutions per site.

**Figure 7 viruses-17-00632-f007:**
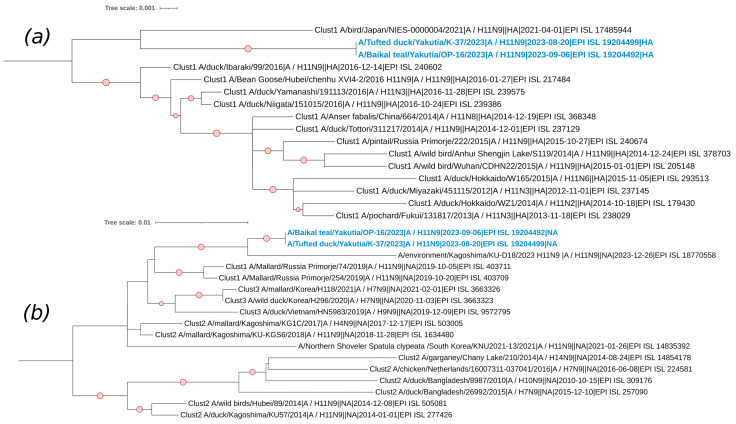
Phylogenetic analysis of H11N9 avian influenza viruses. (**a**) Maximum likelihood phylogenetic tree of the HA (H11) genome segment of avian influenza viruses isolated in the Republic of Sakha (Yakutia). The red circle symbol denotes branches with values bootstrap > 70%. The studied strains collected during this research are highlighted in blue. The tree scale represents the number of substitutions per site. (**b**) Maximum likelihood phylogenetic tree of the NA (N9) genome segment of avian influenza viruses isolated in the Republic of Sakha (Yakutia). The red circle symbol denotes branches with values bootstrap > 70%. The studied strains collected during this research are highlighted in blue. The tree scale represents the number of substitutions per site.

**Figure 8 viruses-17-00632-f008:**
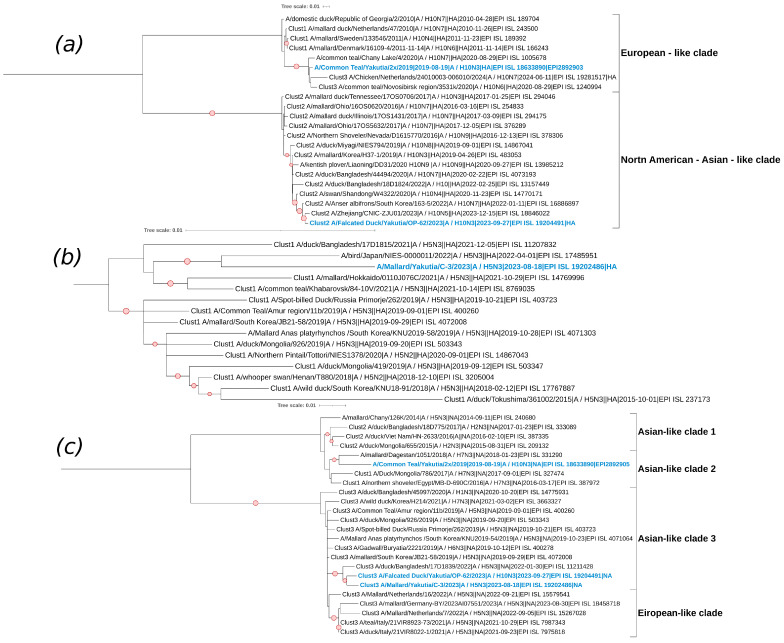
Phylogenetic analysis of H10N3 and H5N3 avian influenza viruses. (**a**) Maximum likelihood phylogenetic tree of the HA (H10) genome segment of avian influenza viruses isolated in the Republic of Sakha (Yakutia). The red circle symbol denotes branches with values bootstrap > 70%. The studied strains collected during this research are highlighted in blue. The tree scale represents the number of substitutions per site. (**b**) Maximum likelihood phylogenetic tree of the HA (H5) genome segment of avian influenza viruses isolated in the Republic of Sakha (Yakutia). The red circle symbol denotes branches with values bootstrap > 70%. The studied strains collected during this research are highlighted in blue. The tree scale represents the number of substitutions per site. (**c**) Maximum likelihood phylogenetic tree of the NA (N3) genome segment of avian influenza viruses isolated in the Republic of Sakha (Yakutia). The red circle symbol denotes branches with values bootstrap > 70%. The studied strains collected during this research are highlighted in blue. The tree scale represents the number of substitutions per site.

**Table 1 viruses-17-00632-t001:** Sample size and results of virus detection in wild waterfowl of Yakutia.

Order	Species	Total Number	Number of AIVs	Positivity Rate (%)
*Accipitriformes* (n = 4)	black kite (*Milvus migrans*)	4	0	0
*Anseriformes* (n = 1851)	greater white-fronted goose (*Anser albifrons*)	13	0	0
	northern pintail (*Anas acuta*)	61	3	4.92
	northern shoveler *(Anas clypeata*)	174	16	9.2
	common teal (*Anas crecca*)	516	36	6.98
	falcated duck (*Anas falcata*)	31	1	3.23
	baikal teal (*Anas formosa*)	20	1	5
	European wigeon (*Anas penelope*)	113	1	0.9
	Mallard (*Anas platyrhynchos*)	220	8	3.63
	garganey (*Anas querquedula*)	58	6	10.34
	snow goose (*Anser caerulescens*)	24	0	0
	bean goose (*Anser fabalis*)	2	0	0
	bar-headed goose (*Anser indicus*)	4	0	0
	common pochard (*Aythya ferina*)	1	0	0
	tufted duck (*Aythya fuligula*)	250	1	0.4
	brant (*Branta bernicla*)	8	0	0
	goldeneye (*Bucephala clangula*)	10	0	0
	long-tailed duck (*Clangula hyemalis*)	9	0	0
	tundra swan (*Cygnus bewickii*)	218	0	0
	harlequin duck (*Histrionicus histrionicus*)	4	0	0
	Eurasian wigeon (*Mareca penelope*)	57	0	0
	velvet scoter (*Melanitta fusca*)	1	0	0
	smew (*Mergellus albellus*)	6	0	0
	common merganser (*Mergus merganser*)	2	0	0
	red-breasted merganser (*Mergus serrator*)	8	0	0
	spectacled eider (*Somateria fischeri*)	41	0	0
*Charadriiformes* (n = 73)	sharp-tailed sandpiper (*Calidris acuminata*)	2	0	0
	pectoral sandpiper (*Calidris melanotos*)	1	0	0
	temminck’s stint (*Calidris temminckii*)	2	0	0
	white-winged tern (*Chlidonias leucopterus*)	4	0	0
	common snipe (*Gallinago gallinago*)	1	0	0
	pintailed snipe (*Gallinago stenura*)	1	0	0
	European herring gull (*Larus argentatus*)	4	0	0
	common gull (*Larus canus*)	10	0	0
	glaucous gull (*Larus hyperboreus*)	1	0	0
	black-headed gull (*Larus ridibundus*)	13	0	0
	vega gull (*Larus vegae*)	4	0	0
	red phalarope (*Phalaropus fulicarius*)	1	0	0
	gray plover (*Pluvialis squatarola*)	1	0	0
	common tern (*Sterna hirundo*)	14	0	0
	wood sandpiper (*Tringa glareola*)	1	0	0
	green shank (*Tringa nebularia*)	1	0	0
	green sandpiper (*Tringa ochropus*)	1	0	0
	marsh sandpiper (*Tringa stagnatilis*)	10	0	0
	northern lapwing (*Vanellus vanellus*)	1	0	0
*Galliformes* (n = 17)	hazel grouse (*Bonasa bonasia*)	7	0	0
	willow ptarmigan (*Lagopus lagopus*)	8	0	0
	black grouse (*Lyrurus tetrix*)	2	0	0
*Gaviiformes* (n = 1)	black-throated loon (*Gavia arctica*)	1	0	0
*Gruiformes* (n = 4)	white-naped crane (*Antigone vipio*)	1	0	0
	coot (*Fulica atra*)	1	0	0
	water rail (*Rallus aquaticus*)	1	0	0
	brown-cheeked rail (*Rallus indicus*)	1	0	0
*Passeriformes* (n = 1)	carrion crow (*Corvus corone*)	1	0	0
*Pelecaniformes* (n = 2)	gray heron (*Ardea cinerea*)	2	0	0
*Podicipediformes* (n = 17)	red-necked grebe (*Podiceps grisegena*)	17	0	0
9 orders	56 species	1970	73	3.71

**Table 2 viruses-17-00632-t002:** The strains of AIV isolated during the study.

Name	Subtype	Host	Date	ID
A/Teal/Yakutia/802/2018	H4N6	common teal (*Anas crecca*)	20 August 2018	EPI ISL 345915
A/Teal/Yakutia/849/2018	H3N8	common teal (*Anas crecca*)	21 August 2018	EPI ISL 345916
A/Common Teal/Yakutia/14/2019	H3N8	common teal (*Anas crecca*)	17 August 2019	EPI ISL 18633896
A/Common Teal/Yakutia/18/2019	H3N8	common teal (*Anas crecca*)	17 August 2019	EPI ISL 18633897
A/Common Teal/Yakutia/19/2019	H3N8	common teal (*Anas crecca*)	17 August 2019	EPI ISL 18633898
A/Common Teal/Yakutia/32/2019	H3N8	common teal (*Anas crecca*)	17 August 2019	EPI ISL 18633899
A/Northern pintail/Yakutia/41/2019	H3N8	northern pintail (*Anas acuta*)	18 August 2019	EPI ISL 18633900
A/Shoveler/Yakutia/56/2019	H3N8	northern shoveler (*Anas clypeata*)	18 August 2019	EPI ISL 18633901
A/Shoveler/Yakutia/61/2019	H3N8	northern shoveler (*Anas clypeata*)	18 August 2019	EPI ISL 18633902
A/Common Teal/Yakutia/2x/2019	H10N3	common teal (*Anas crecca*)	19 August 2019	EPI ISL 18633890
A/Common Teal/Yakutia/6x/2019	H4N6	common teal (*Anas crecca*)	19 August 2019	EPI ISL 18633892
A/Shoveler/Yakutia/28x/2019	H4N6	northern shoveler (*Anas clypeata*)	19 August 2019	EPI ISL 18633893
A/Common Teal/Yakutia/63x/2019	H3N8	common teal (*Anas crecca*)	20 August 2019	EPI ISL 18633895
A/Shoveler/Yakutia/68x/2019	H4N6	northern shoveler (*Anas clypeata*)	20 August 2019	EPI ISL 18633894
A/Shoveler/Yakutia/57/2019	H4N6	northern shoveler (*Anas clypeata*)	8 September 2018	EPI ISL 18633935
A/Common Teal/Yakutia/49/2020	H3N8	common teal (*Anas crecca*)	13 August 2020	EPI ISL 1081354
A/European Wigeon/Yakutia/55/2020	H3N8	European wigeon (*Anas penelope*)	13 August 2020	EPI ISL 1081355
A/Mallard/Yakutia/11/2020	H3N8	mallard (*Anas platyrhynchos*)	13 August 2020	EPI ISL 1081356
A/Mallard/Yakutia/21/2020	H3N8	mallard (*Anas platyrhynchos*)	13 August 2020	EPI ISL 1081357
A/Mallard/Yakutia/47/2020	H7N7	mallard (*Anas platyrhynchos*)	13 August 2020	EPI ISL 1081358
A/Northern Pintail/Yakutia/14/2020	H3N8	northern pintail (*Anas acuta*)	13 August 2020	EPI ISL 1081359
A/Shoveler/Yakutia/25/2020	H3N8	northern shoveler (*Anas clypeata*)	13 August 2020	EPI ISL 1081360
A/Common Teal/Yakutia/128/2020	H3N8	common teal (*Anas crecca*)	14 August 2020	EPI ISL 1081362
A/Garganey/Yakutia/94/2020	H3N8	garganey (*Anas querquedula*)	14 August 2020	EPI ISL 1081366
A/Garganey/Yakutia/98/2020	H3N8	garganey (*Anas querquedula*)	14 August 2020	EPI ISL 1081367
A/Shoveler/Yakutia/65/2020	H3N8	northern shoveler (*Anas clypeata*)	14 August 2020	EPI ISL 1081361
A/Common Teal/Yakutia/57x/2020	H3N8	common teal (*Anas crecca*)	15 August 2020	EPI ISL 1081363
A/Common Teal/Yakutia/59x/2020	H3N8	common teal (*Anas crecca*)	15 August 2020	EPI ISL 1081364
A/Common Teal/Yakutia/60x/2020	H3N8	common teal (*Anas crecca*)	15 August 2020	EPI ISL 1081365
A/Common Teal/Yakutia/13/2021	H3N8	common teal (*Anas crecca*)	14 August 2021	EPI ISL 19212486
A/Common Teal/Yakutia/18/2021	H3N6	common teal (*Anas crecca*)	15 August 2021	EPI ISL 19212478
A/Common Teal/Yakutia/80/2021	H3N8	common teal (*Anas crecca*)	15 August 2021	EPI ISL 19212539
A/Common Teal/Yakutia/111/2021	H3N8	common teal (*Anas crecca*)	21 August 2021	EPI ISL 19212481
A/Common Teal/Yakutia/122/2021	H3N8	common teal (*Anas crecca*)	21 August 2021	EPI ISL 19212528
A/Common Teal/Yakutia/146/2021	H4N6	common teal (*Anas crecca*)	22 August 2021	EPI ISL 19212487
A/Common Teal/Yakutia/148/2021	H3N6	common teal (*Anas crecca*)	22 August 2021	EPI ISL 19212484
A/Mallard/Yakutia/178/2021	H3N6	mallard (*Anas platyrhynchos*)	22 August 2021	EPI ISL 19212482
A/Mallard/Yakutia/179/2021	H3N6	mallard (*Anas platyrhynchos*)	22 August 2021	EPI ISL 19212483
A/Shoveler/Yakutia/141/2021	H3N8	northern shoveler (*Anas clypeata*)	22 August 2021	EPI ISL 19212529
A/MallardxNorthern Pintail Hybrid/Yakutia/V46/2022	H4N6	hybrid between garganey (*Anas querquedula*) and northern pintail (*Anas acuta*)	15 August 2022	EPI ISL 19212538
A/Shoveler/Yakutia/Bo17/2022	H4N6	northern shoveler (*Anas clypeata*)	20 August 2022	EPI ISL 19212537
A/Shoveler/Yakutia/Bo29/2022	H4N6	northern shoveler (*Anas clypeata*)	20 August 2022	EPI ISL 19212536
A/Shoveler/Yakutia/Bo30/2022	H4N6	northern shoveler (*Anas clypeata*)	20 August 2022	EPI ISL 19212535
A/Common Teal/Yakutia/V98/2022	H3N8	common teal (*Anas crecca*)	24 August 2022	EPI ISL 19212534
A/Common Teal/Yakutia/V109/2022	H4N6	common teal (*Anas crecca*)	5 September 2022	EPI ISL 19212533
A/Mallard/Yakutia/C-3/2023	H5N3	mallard (*Anas platyrhynchos*)	18 August 2023	EPI ISL 19202486
A/Mallard/Yakutia/C-9/2023	H4N6	mallard (*Anas platyrhynchos*)	18 August 2023	EPI ISL 19202488
A/Common Teal/Yakutia/C-52/2023	H3N6	common teal (*Anas crecca*)	19 August 2023	EPI ISL 19202490
A/Common Teal/Yakutia/K-28/2023	H3N8	common teal (*Anas crecca*)	19 August 2023	EPI ISL 19204497
A/Shoveler/Yakutia/K-17/2023	H3N8	northern shoveler (*Anas clypeata*)	19 August 2023	EPI ISL 19204496
A/Shoveler/Yakutia/K-33/2023	H3N8	northern shoveler (*Anas clypeata*)	19 August 2023	EPI ISL 19204498
A/Anas crecca/Yakutia/K-57/2023	H3N8	common teal (*Anas crecca*)	20 August 2023	EPI ISL 19204504
A/Common Teal/Yakutia/C-67/2023	H3N8	common teal (*Anas crecca*)	20 August 2023	EPI ISL 19202492
A/Common Teal/Yakutia/C-74/2023	H3N8	common teal (*Anas crecca*)	20 August 2023	EPI ISL 19202494
A/Common Teal/Yakutia/C-83/2023	H3N8	common teal (*Anas crecca*)	20 August 2023	EPI ISL 19202498
A/Common Teal/Yakutia/C-85/2023	H3N8	common teal (*Anas crecca*)	20 August 2023	EPI ISL 19202501
A/Common Teal/Yakutia/C-86/2023	H3N8	common teal (*Anas crecca*)	20 August 2023	EPI ISL 19202503
A/Garganey/Yakutia/K-74/2023	H3N8	garganey (*Anas querquedula*)	20 August 2023	EPI ISL 19204503
A/Garganey/Yakutia/K-75/2023	H3N8	garganey (*Anas querquedula*)	20 August 2023	EPI ISL 19204502
A/Mallard/Yakutia/K-52/2023	H3N8	mallard (*Anas platyrhynchos*)	20 August 2023	EPI ISL 19204493
A/Northern Pintail/Yakutia/C-82/2023	H3N8	northern pintail (*Anas acuta*)	20 August 2023	EPI ISL 19202496
A/Northern Shoveler/Yakutia/K-56/2023	H4N6	northern shoveler (*Anas clypeata*)	20 August 2023	EPI ISL 19204505
A/Shoveler/Yakutia/K-68/2023	H3N8	northern shoveler (*Anas clypeata*)	20 August 2023	EPI ISL 19204494
A/Shoveler/Yakutia/K-72/2023	H4N6	northern shoveler (*Anas clypeata*)	20 August 2023	EPI ISL 19204495
A/Tufted Duck/Yakutia/K-37/2023	H11N9	tufted duck (*Aythya fuligula*)	20 August 2023	EPI ISL 19204499
A/Common Teal/Yakutia/K-96/2023	H4N6	common teal (*Anas crecca*)	27 August 2023	EPI ISL 19204501
A/Common Teal/Yakutia/K-98/2023	H3N8	common teal (*Anas crecca*)	27 August 2023	EPI ISL 19204500
A/Common Teal/Yakutia/UA-3/2023	H3N6	common teal (*Anas crecca*)	30 August 2023	EPI ISL 19202504
A/Common Teal/Yakutia/UA-4/2023	H3N8	common teal (*Anas crecca*)	30 August 2023	EPI ISL 19202505
A/Common Teal/Yakutia/UA-37/2023	H3N6	common teal (*Anas crecca*)	3 September 2023	EPI ISL 19204506
A/Common Teal/Yakutia/UA-52/2023	H3N8	common teal (*Anas crecca*)	5 September 2023	EPI ISL 19204489
A/Common Teal/Yakutia/UA-55/2023	H3N6	common teal (*Anas crecca*)	5 September 2023	EPI ISL 19204490
A/Baikal Teal/Yakutia/OP-16/2023	H11N9	baikal teal (*Anas formosa)*	6 September 2023	EPI ISL 19204492
A/Falcated Duck/Yakutia/OP-62/2023	H10N3	falcated duck (*Anas falcata*)	27 September 2023	EPI ISL 19204491

**Table 3 viruses-17-00632-t003:** The affiliation of each segment of the isolated AIV strains to a specific cluster is reflected by color. Each segment is assigned its own color: PB2 is yellow, PB1 is gray, PA is red, NP is purple, MP is blue, and NS is green. The cluster affiliation is indicated by a shade of the respective color (e.g., shades of yellow for PB2, shades of gray for PB1, and so on). Strains with identical segment sets based on cluster affiliation are divided into provisional genotypes, as shown in the Genotype column, labeled as g1, g2, etc.

Virus	Subtype	PB2	PB1	PA	NP	MP	NS	Genotype
802/2018	H4N6	Mongolia	Shanghai	Mongolia	Korea	Bangladesh	Shandong	
148/2021	H3N6	Vietnam	South_Korea	Chany	Jiangxi	Hunan	Mongolia	
47/2020	H7N7	Buryatia			South_Korea	Korea	Kagoshima	
UA-4/2023	H3N8	Hubei	Amur_region		Shandong	Hunan	Novosibirsk_region	
C-82/2023	H3N8	Russia_Primorje	Russia_Primorje	Amur_region	Mongolia	Mongolia	Russia_Primorje	
C-83/2023	H3N8						Mongolia	
2x/2019	H10N3	Omsk_Region	Ningxia	Chany	Tomsk	Korea	Germany-HE	
C-3/2023	H5N3	Bangladesh	Saga	North_Kazakhstan	Korea	Bangladesh	Novosibirsk_region	
UA-3/2023	H3N6	Russia_Primorje	Amur_region	Amur_region	Yamaguchi	Shanxi	Jiangxi	
UA-52/2023	H3N8			Chany		Mongolia	Mongolia	
V98/2022	H3N8		Toyama	Kagoshima	Alaska		South_Africa	
K-52/2023	H3N8		Kagoshima		Jiangxi		Mongolia	
179/2021	H3N6	Shandong	Saga	Amur_region		Hokkaido	Amur_region	
18/2019	H3N8	Mongolia	Mongolia	Korea	Ibaraki	Korea	Hunan	
122/2021	H3N8	Shandong	Korea	South_Korea	Jiangxi	Georgia	Mongolia	
V109/2022	H4N6		South_Korea	Japan	Wakayama	Zhejiang	South_Korea	
OP-62/2023	H10N3			Bangladesh	South_Korea	Bangladesh	Jiangxi	
V46/2022	H4N6				Shandong	Hokkaido	Hubei	
13/2021	H3N8		Toyama	Chiba	Jiangxi	Bangladesh	Mongolia	
C-52/2023	H3N6		Kagoshima	Bangladesh	South_Korea	Bangladesh	Mongolia	
111/2021	H3N8		Toyama	South_Korea	Jiangxi	Hokkaido	South_Africa	
UA-55/2023	H3N6		Bangladesh	Amur_region			Amur_region	
Bo30/2022	H4N6	Mongolia	Sakhalin	South_Korea	South_Korea	Bangladesh	Mongolia	
UA-37/2023	H3N6		Amur_region	Amur_region	Jiangxi	Hokkaido	Amur_region	
18/2021	H3N6	Shandong	Korea	Bangladesh	Jiangxi	Mongolia	Kagoshima	
63x/2019	H3N8	Mongolia	Mongolia	Amur_region	Kyoto	Korea	Hunan	
41/2019	H3N8		Mongolia	Korea	Ibaraki			
178/2021	H3N6		Toyama	Amur_region	Jiangxi	Hokkaido	Amur_region	
14/2019	H3N8	Egypt	Mongolia	Korea	Ibaraki	Hunan	Shandong	g1
32/2019	H3N8							
56/2019	H3N8							
61/2019	H3N8							
849/2018	H3N8	Novosibirsk_region						
K-57/2023	H3N8	Bangladesh	Shandong	Chany	Mongolia	Shanxi	Buryatia	g2
K-74/2023	H3N8							
K-33/2023	H3N8			Bangladesh		Bangladesh	Kagoshima	
146/2021	H4N6	Shandong	Toyama	Bangladesh	Jiangxi	Mongolia	Egypt	
C-67/2023	H3N8	South_Korea			Bangladesh		Mongolia	
C-85/2023	H3N8		Kagoshima	Okayama		Bangladesh	Mongolia	
C-9/2023	H4N6	Shandong	Toyama		Jiangxi	Mongolia	Omsk_region	
K-28/2023	H3N8	South_Korea	Kagoshima	Bangladesh	Yamaguchi	Bangladesh	Kagoshima	g3
K-96/2023	H4N6	Shandong		Kagoshima	Jiangxi	Mongolia		
K-98/2023	H3N8	South_Korea		Bangladesh		Bangladesh	South_Korea	
K-56/2023	H4N6	Shandong	Toyama	Kagoshima				
Bo17/2022	H4N6					Mongolia	Shanghai	
Bo29/2022	H4N6					Bangladesh	Kagoshima	
K-17/2023	H3N8	South_Korea	Kagoshima	Bangladesh			South_Korea	
K-68/2023	H3N8	Shandong	Bangladesh	Okayama			Kagoshima	
K-72/2023	H4N6		Toyama	Kagoshima		Mongolia	Buryatia	
OP-16/2023	H11N9	Kagoshima	Kagoshima	Hokkaido	South_Korea	Tottori	South_Korea	g4
K-37/2023	H11N9							
80/2021	H3N8	Shandong	Toyama	South_Korea	Jiangxi	Hokkaido	Alaska	g5
141/2021	H3N8						Mongolia	
C-74/2023	H3N8	Shandong	Bangladesh	Mongolia	Shandong	Hokkaido	Mongolia	g6
C-86/2023	H3N8		Kagoshima		Bangladesh			
K-75/2023	H3N8	Kagoshima	Bangladesh		Jiangxi		Shimane	
128/2020	H3N8	Mongolia	Mongolia	Amur_region	Ibaraki	Korea	Shandong	g7
49/2020	H3N8							
57x/2020	H3N8							
59x/2020	H3N8							
60x/2020	H3N8							
55/2020	H3N8							
94/2020	H3N8							
98/2020	H3N8							
11/2020	H3N8							
21/2020	H3N8							
14/2020	H3N8							
25/2020	H3N8							
65/2020	H3N8							
19/2019	H3N8	Mongolia	Mongolia	Amur_region	Eastern_China	Korea	Hunan	g8
57/2019	H4N6							
68x/2019	H4N6							
6x/2019	H4N6	Mongolia	Mongolia	Amur_region	Eastern_China	Korea	Hunan	g9
28x/2019	H4N6							

**Table 4 viruses-17-00632-t004:** Amino acid substitutions found in the studied strains, according to Suttie et al.

Segment/Protein	Mutation	Effects	Strains	References
PB2	I292V	Increased polymerase activity in mammalian cell line, increased virulence in mice Increased polymerase activity in mammalian cell line	A/Common Teal/Yakutia/2x/2019 (H10N3) A/Mallard/Yakutia/C-3/2023 (H5N3) A/Common Teal/Yakutia/148/2021 (H3N6) A/Garganey/Yakutia/K-74/2023 (H3N8) A/Shoveler/Yakutia/K-33/2023 (H3N8) A/Anas crecca/Yakutia/K-57/2023 (H3N8)	[26]
PB2	K482R	Increased polymerase activity in mammalian cell line	A/Common Teal/Yakutia/UA-37/2023 (H3N6)	[27,28]
PB2	A588V	Increased polymerase activity and replication in mammalian and avian cell lines, increased virulence in mice	A/Common Teal/Yakutia/148*/2021 (H3N6)	[29]
PB1-F2	N66S	Enhanced replication, virulence, and antiviral response in mice	A/Teal/Yakutia/849/2018 (H3N8) A/Mallard/Yakutia/47/2020 (H7N7) A/Shoveler/Yakutia/56/2019 (H3N8) A/Shoveler/Yakutia/61/2019 (H3N8) A/Northern Pintail/Yakutia/14/2020(H3N8) A/Common Teal/Yakutia/18/2019 (H3N8) A/Common Teal/Yakutia/32/2019 (H3N8) A/Common Teal/Yakutia/63x/2019 (H3N8) A/Common Teal/Yakutia/2x/2019 (H10N3) A/Common Teal/Yakutia/V109/2022(H4N6) A/Common Teal/Yakutia/122/2021 (H3N8) A/Common Teal/Yakutia/18/2021 (H3N6) A/MallardxNorthern Pintail Hybrid/Yakutia/V46/2022 (H4N6) A/Common Teal/Yakutia/148/2021 (H3N6) A/Falcated Duck/Yakutia/OP-62/2023 (H10N3) A/Garganey/Yakutia/K-74/2023 (H3N8) A/Shoveler/Yakutia/K-33/2023 (H3N8) A/Anas crecca/Yakutia/K-57/2023 (H3N8)	[30,31]
PA	K356R	Increase polymerase activity and enhanced replication in mammalian cell line, increased virulence in mice	A/Common Teal/Yakutia/C-67/2023(H3N8) A/Common Teal/Yakutia/C-52/2023(H3N6) A/Common Teal/Yakutia/C-85/2023(H3N8)	[32]
PA	S224P, N383D	Increased polymerase activity and enhanced viral replication in duck and mouse cell lines, increased virulence in mice and ducks	A/Common Teal/Yakutia/80/2021 (H3N8)	[33,34]
NP	M105V	Increased virulence in chickens	A/Mallard/Yakutia/47/2020 (H7N7) A/Common Teal/Yakutia/2x/2019 (H10N3) A/Shoveler/Yakutia/Bo30/2022 (H4N6) A/Common Teal/Yakutia/C-52/2023(H3N6) A/Common Teal/Yakutia/UA-4/2023 (H3N8) A/Falcated Duck/Yakutia/OP-62/2023 (H10N3)	[35,36]
NP	I109T	Increased polymerase activity and viral replication in chickens (but not ducks), increased virulence in chickens	A/Common Teal/Yakutia/UA-4/2023 (H3N8)	[35,36]

## Data Availability

All genome sequences of AIVs from this study are available in GISAID database (accession numbers: EPI ISL 1081354, EPI ISL 1081355, EPI ISL 1081356, EPI ISL 1081357, EPI ISL 1081358, EPI ISL 1081359, EPI ISL 1081360, EPI ISL 1081361, EPI ISL 1081362, EPI ISL 1081363, EPI ISL 1081364, EPI ISL 1081365, EPI ISL 1081366, EPI ISL 1081367, EPI ISL 18633890, EPI ISL 18633892, EPI ISL 18633893, EPI ISL 18633894, EPI ISL 18633895, EPI ISL 18633896, EPI ISL 18633897, EPI ISL 18633898, EPI ISL 18633899, EPI ISL 18633900, EPI ISL 18633901, EPI ISL 18633902, EPI ISL 18633935, EPI ISL 19202486, EPI ISL 19202488, EPI ISL 19202490, EPI ISL 19202492, EPI ISL 19202494, EPI ISL 19202496, EPI ISL 19202498, EPI ISL 19202501, EPI ISL 19202503, EPI ISL 19202504, EPI ISL 19202505, EPI ISL 19204489, EPI ISL 19204490, EPI ISL 19204491, EPI ISL 19204492, EPI ISL 19204493, EPI ISL 19204494, EPI ISL 19204495, EPI ISL 19204496, EPI ISL 19204497, EPI ISL 19204498, EPI ISL 19204499, EPI ISL 19204500, EPI ISL 19204501, EPI ISL 19204502, EPI ISL 19204503, EPI ISL 19204504, EPI ISL 19204505, EPI ISL 19204506, EPI ISL 19212478, EPI ISL 19212481, EPI ISL 19212482, EPI ISL 19212483, EPI ISL 19212484, EPI ISL 19212486, EPI ISL 19212487, EPI ISL 19212528, EPI ISL 19212529, EPI ISL 19212533, EPI ISL 19212534, EPI ISL 19212535, EPI ISL 19212536, EPI ISL 19212537, EPI ISL 19212538, EPI ISL 19212539, EPI ISL 345915, and EPI ISL 345916).

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
