# Peer review of "Exploring Avian Influenza Viruses in Yakutia—The Largest Breeding Habitat of Wild Migratory Birds in Northeastern Siberia"

_viruses, 2025, doi:10.3390/v17050632_

Round 1
Reviewer 1 Report
Comments and Suggestions for Authors
This study systematically reveals the crucial role of the Yakutia region as a migratory hub for Eurasian birds in the transmission of avian influenza virus (AIV). Through multi - year monitoring, the genetic diversity of seven AIV subtypes (such as H3N6 and H10N3) has been discovered, along with their associations with virus strains from East Asia, Europe, and the Americas. It has also been confirmed that there is a stable circulation of virus genotypes and mammalian - adapted mutations in this region, providing important data support for global research on the ecological evolution of AIV and early warning of cross - species transmission risks. However, there are still some parts of this study that need to be appropriately modified:
- Although the sample size and isolation number were provided in table 1, however, it is not clear to show the surveillance in Yakutia region. It is recommended to integrate the sample information, time, separation rate, etc., and present them in the form of a line chart or a bar chart.
- The map of the sampling sites in figure 1 should be improved, some information including the migratory flyway, sample number in each sites, and the field photo can be integrated together in figure 1.
- When elaborating on the importance and current status of AIV research in the Yakutia regionof EAA flyway, the contributions of existing studies should be fully mentioned. Due to large portion of H3N6 and H3N8 viruses were identified, the previous study (PMID: 36913241) conducted an in - depth investigation into the H3N8 subtype AIV carried by migratory birds in eastern China from 2017 to 2021 should be compared and discussed. Moreover, the H7, H10 and H11 subtypes identified in this study should be compared with the H7, H10 and H11 viruses founded in Shandong, eastern China (PMID: 39248597; PMID: 35389243; PMID: 35580801). Appropriate citation of this study can provide a broader perspective for exploring AIV in EAA flyway, including (Yakutia region and eastern China) and help us understand the transmission and evolution patterns of AIV across different regions and hosts.
- In line 35 of the abstract, the full name of "VAM" is not defined. It is recommended to supplement the full name and its definition when it first appears.
- In this study, only HA - positive samples were tested for the AIV M gene by PCR, and it is not explained whether other tests were performed on HA - negative samples to rule out AIV infection. This may lead to the omission of some virus - positive samples. It is recommended to supplement relevant test information or explain the reason for not conducting such tests.
- In Tables 1 and 2, some data formats are inconsistent. For example, the numbers in "Positivity rate" use commas as decimal separators (such as 4,92), which does not conform to the scientific data representation method. It is recommended to uniformly change them to decimal points (such as 4.92).
- In the "Species" column of Table 1, the word "Gul" in "black - headed Gul (Larus ridibundusl)" is misspelled and should be "Gull".
The language can be improved.
Author Response
Dear Reviewers, Editor,
Thank you all for your useful comments and constructive suggestions which helped us to improve the manuscript. Additional information was provided for results and discussion according to the comments. We edited/modified study and text significantly according to comments and suggestions.
Please see below detailed response on each point of the review and corrections of the manuscript.
Reviewer 1
This study systematically reveals the crucial role of the Yakutia region as a migratory hub for Eurasian birds in the transmission of avian influenza virus (AIV). Through multi - year monitoring, the genetic diversity of seven AIV subtypes (such as H3N6 and H10N3) has been discovered, along with their associations with virus strains from East Asia, Europe, and the Americas. It has also been confirmed that there is a stable circulation of virus genotypes and mammalian - adapted mutations in this region, providing important data support for global research on the ecological evolution of AIV and early warning of cross - species transmission risks. However, there are still some parts of this study that need to be appropriately modified:
1. Although the sample size and isolation number were provided in table 1, however, it is not clear to show the surveillance in Yakutia region. It is recommended to integrate the sample information, time, separation rate, etc., and present them in the form of a line chart or a bar chart.
Response: We agree with the reviewer’s comment. To avoid overloading the manuscript, we have included comprehensive surveillance data in Table S 1, Sheet 2 of the supplementary Excel spreadsheet. Since our data collection occurs exclusively during the official hunting period each year, we are unable to provide information on year-round monitoring or build continuous schedules throughout the entire year. Consequently, our findings pertain primarily to a temporary cut in circulation during the autumn period, coinciding with the onset of bird migration to wintering grounds.
We have clarified this point by adding specific details to line 94 and have also included this limitation in the discussion section for transparency.
2. The map of the sampling sites in figure 1 should be improved, some information including the migratory flyway, sample number in each sites, and the field photo can be integrated together in figure 1.
Response: Thank you for this fine suggestion! We tried to integrate suggested information to the map, we tried to combine the general flyways scheme (Veen et al., 2005). The sample number in each site – we would like to keep it in the Table S1, which was modified according to the previous comment.
3. When elaborating on the importance and current status of AIV research in the Yakutia region of EAA flyway, the contributions of existing studies should be fully mentioned. Due to large portion of H3N6 and H3N8 viruses were identified, the previous study (PMID: 36913241) conducted an in - depth investigation into the H3N8 subtype AIV carried by migratory birds in eastern China from 2017 to 2021 should be compared and discussed. Moreover, the H7, H10 and H11 subtypes identified in this study should be compared with the H7, H10 and H11 viruses founded in Shandong, eastern China (PMID: 39248597; PMID: 35389243; PMID: 35580801). Appropriate citation of this study can provide a broader perspective for exploring AIV in EAA flyway, including (Yakutia region and eastern China) and help us understand the transmission and evolution patterns of AIV across different regions and hosts.
Response: We absolutely agree with a comment. We significantly modified the discussion section according to the suggestions, added valuable references.
4. In line 35 of the abstract, the full name of "VAM" is not defined. It is recommended to supplement the full name and its definition when it first appears.
Response: corrected according to the comment
5. In this study, only HA - positive samples were tested for the AIV M gene by PCR, and it is not explained whether other tests were performed on HA - negative samples to rule out AIV infection. This may lead to the omission of some virus - positive samples. It is recommended to supplement relevant test information or explain the reason for not conducting such tests.
Response: There are a number of studies comparing the virus isolation method on chicken embryos and PCR. A number of works indicate a greater sensitivity of PCR diagnostics (for example, Kim et al., 2019, doi: 10.4142/jvs.2019.20.e56), others indicate a higher sensitivity of isolation in chicken embryos (for example, Spackman et al., 2003, DOI: 10.1637/0005-2086-47.s3.1079). The results on average show that they can be compared in sensitivity. Obviously, separate methods of detection have some limitations. Therefore, different methods should be combined for optimal surveillance. We choose VI method with further PCR and sequencing.
We conducted three consecutive inoculation passages in specific pathogen-free (SPF) chicken embryos. Samples demonstrating no hemagglutinating activity after these passages were considered as negative.
We added specific detail about three passages.
6. In Tables 1 and 2, some data formats are inconsistent. For example, the numbers in "Positivity rate" use commas as decimal separators (such as 4,92), which does not conform to the scientific data representation method. It is recommended to uniformly change them to decimal points (such as 4.92).
Response: changed according to the comment
7. In the "Species" column of Table 1, the word "Gul" in "black - headed Gul (Larus ridibundusl)" is misspelled and should be "Gull".
Response: changed according to the comment
Reviewer 2 Report
Comments and Suggestions for Authors
I would like to congratulate all researchers for their efforts to produce this interesting and valuable work on the epidemiology of Avian influenza viruses in wild birds.
I have only few minor comments
Line 50: add ruminants ((dairy cattle in the USA , ref: Sreenivasan et al.2024, Burrough et al.2024..)
Line 63-64 : Okazaki et al.2000
Line 70 : Marchenkoet al 2015
Line 94: add the date of hunting period ( from....to.....)
Author Response
Dear Reviewers, Editor,
Thank you all for your useful comments and constructive suggestions which helped us to improve the manuscript. Additional information was provided for results and discussion according to the comments. We edited/modified study and text significantly according to comments and suggestions.
Please see below detailed response on each point of the review and corrections of the manuscript.
Reviewer 2
I would like to congratulate all researchers for their efforts to produce this interesting and valuable work on the epidemiology of Avian influenza viruses in wild birds.
I have only few minor comments
Line 50: add ruminants ((dairy cattle in the USA , ref: Sreenivasan et al.2024, Burrough et al.2024..)
Response: Thanks for appreciating our study. changed according to the comment
Line 63-64 : Okazaki et al.2000
Response: changed according to the comment
Line 70 : Marchenkoet al 2015
Response: changed according to the comment
Line 94: add the date of hunting period ( from....to.....)
Response: we added information according to the comment
Reviewer 3 Report
Comments and Suggestions for Authors
The manuscript by Kasianov et al. investigates the ecology and evolution of avian influenza viruses (AIV) in Yakutia, the largest breeding habitat for wild migratory birds in Northeastern Siberia. The study provides valuable insights into the genetic diversity, phylogenetic relationships, and potential risks associated with AIV in this ecologically significant region. Key findings include the identification of multiple AIV subtypes, predominantly within the Eurasian lineage, and the detection of amino acid substitutions linked to increased virulence and mammalian adaptation. Additionally, the research highlights Yakutia’s role as a hub for viral exchange along migratory bird routes. While the study offers promising results and contributes to the understanding of AIV dynamics in an understudied region, several shortcomings need to be addressed to enhance the clarity and robustness of the conclusions.
Major issues:
- The manuscript does not sufficiently justify the selection of Yakutia as the study region. While it mentions the ecological importance of Yakutia, it lacks a detailed description of the region's unique ecological features and how they influence AIV transmission and evolution. The authors should provide more background information on Yakutia's ecological characteristics and its role in global AIV spread.
- The study includes 93.96% of samples from Anseriformes, which may introduce bias. The authors should justify the sampling strategy and discuss its impact on the results. Additionally, the distribution of samples across different years and seasons is unclear. A detailed breakdown of annual sample sizes and an analysis of how seasonality influences virus detection rates are necessary.
- Table 3 uses color coding for gene clusters but doesn't explain what the colors mean. Also, the phylogenetic trees in Figures 2-8 are quite blurry, making it hard to see details. Improving the figure resolution and clarifying the color coding would enhance clarity.
- The manuscript mentions the detection of amino acid substitutions associated with increased virulence and mammalian adaptation but does not explore their implications thoroughly. The authors should provide a more detailed discussion of how these substitutions might affect virus behavior and potential interspecies transmission risks, including their prevalence in other AIVs and relevance to mammalian adaptation.
Minor issues:
- Table 4 lists several amino acid substitutions but does not validate their effects using experimental data from this study. The authors should correlate these substitutions with phenotypic data from in vitro or in vivo experiments to strengthen the findings.
- The manuscript should include more detailed statistical analyses to support the distribution of different AIV subtypes and the significance of the findings.
- The manuscript would benefit from improved readability through simplified sentence structures and enhanced logical flow, particularly in the discussion section where results should be explicitly connected to the study's hypotheses and broader implications.
Author Response
Dear Reviewers, Editor,
Thank you all for your useful comments and constructive suggestions which helped us to improve the manuscript. Additional information was provided for results and discussion according to the comments. We edited/modified study and text significantly according to comments and suggestions.
Please see below detailed response on each point of the review and corrections of the manuscript.
Reviewer 3
The manuscript by Kasianov et al. investigates the ecology and evolution of avian influenza viruses (AIV) in Yakutia, the largest breeding habitat for wild migratory birds in Northeastern Siberia. The study provides valuable insights into the genetic diversity, phylogenetic relationships, and potential risks associated with AIV in this ecologically significant region. Key findings include the identification of multiple AIV subtypes, predominantly within the Eurasian lineage, and the detection of amino acid substitutions linked to increased virulence and mammalian adaptation. Additionally, the research highlights Yakutia’s role as a hub for viral exchange along migratory bird routes. While the study offers promising results and contributes to the understanding of AIV dynamics in an understudied region, several shortcomings need to be addressed to enhance the clarity and robustness of the conclusions.
Major issues:
1. The manuscript does not sufficiently justify the selection of Yakutia as the study region. While it mentions the ecological importance of Yakutia, it lacks a detailed description of the region's unique ecological features and how they influence AIV transmission and evolution. The authors should provide more background information on Yakutia's ecological characteristics and its role in global AIV spread.
Response: We agree with this serious comment. We added information on the ecological features of the Yakutian region. We modified the discussion section.
2. The study includes 93.96% of samples from Anseriformes, which may introduce bias. The authors should justify the sampling strategy and discuss its impact on the results. Additionally, the distribution of samples across different years and seasons is unclear. A detailed breakdown of annual sample sizes and an analysis of how seasonality influences virus detection rates are necessary.
Response: We agree with the reviewer’s comment. Thank you for your helpful comments. To avoid overloading the main manuscript, we have included comprehensive surveillance data in Table S1, Sheet 2 of the supplementary Excel spreadsheet.
Since our data collection is limited to the official hunting season each year, we are unable to provide information on year-round monitoring or establish continuous surveillance schedules throughout the entire year. As a result, our findings primarily reflect a temporary reduction in virus circulation during the autumn period, which coincides with the onset of bird migration to wintering grounds. This approach is consistent with other surveillance programs that focus sampling during hunting seasons, which may influence the observed prevalence and dynamics of avian influenza viruses25.
We have clarified this limitation by adding specific details to line 94 and have also included it in the discussion section for transparency. In the limitations section, we note that 93.96% of our samples are from Anseriformes, which may introduce bias and influence the results. However, we focus on this major reservoir of avian influenza viruses to assess the general diversity of circulating viruses, as Anseriformes are recognized as key hosts in the epidemiology of avian influenza
3. Table 3 uses color coding for gene clusters but doesn't explain what the colors mean. Also, the phylogenetic trees in Figures 2-8 are quite blurry, making it hard to see details. Improving the figure resolution and clarifying the color coding would enhance clarity.
Response: We added some information to the Legend and corrected according to the comment.
4. The manuscript mentions the detection of amino acid substitutions associated with increased virulence and mammalian adaptation but does not explore their implications thoroughly. The authors should provide a more detailed discussion of how these substitutions might affect virus behavior and potential interspecies transmission risks, including their prevalence in other AIVs and relevance to mammalian adaptation.
Response: We identified amino acid substitutions associated with increased virulence and adaptation in mammals. These substitutions can influence viral behavior and the potential risk of interspecies transmission, highlighting their prevalence in avian influenza viruses (AIVs) and their relevance to mammalian adaptation. However, given the environmental context and the minimal likelihood of contact with domestic animals and humans, we consider the potential public health impact to be low. Nevertheless, the detection of novel viruses and their amino acid substitutions in natural populations of wild migratory birds is a critical finding that warrants publication and dissemination as part of early warning efforts. We added these thoughts to the discussion section. However, we avoid very detailed analysis of substitutions, since this is not confirmed by our experimental data and this was not the purpose of this study.
Minor issues:
1. Table 4 lists several amino acid substitutions but does not validate their effects using experimental data from this study. The authors should correlate these substitutions with phenotypic data from in vitro or in vivo experiments to strengthen the findings.
Response: Please refer to our previous comment. We have intentionally avoided a detailed analysis of amino acid substitutions, as such interpretations are not supported by our experimental data and were beyond the scope of this study. We did not conduct experiments specifically focused on substitution effects, including reverse genetics, which are necessary for such investigations. Instead, we believe that our ecological and phylogenetic analyses provide valuable and important insights for this research area. We put it also in Limitation section.
2. The manuscript should include more detailed statistical analyses to support the distribution of different AIV subtypes and the significance of the findings.
Response: We have deliberately refrained from conducting a more detailed statistical analysis on the distribution of different AIV subtypes for the reasons outlined in the limitations section. Furthermore, despite the large sample size, the number of influenza viruses representing subtypes other than H3N8 is too small for robust statistical analysis.
3. The manuscript would benefit from improved readability through simplified sentence structures and enhanced logical flow, particularly in the discussion section where results should be explicitly connected to the study's hypotheses and broader implications.
Response: Thank you for reading the manuscript very carefully. We tried to modified the text, particularly the discussion section according to the comment. If necessary, we would like to apply for English editing.
Sincerely,
Authors